# FABM-PCLake – linking aquatic ecology with hydrodynamics

**F. Hu[1,2*], K. Bolding[1,3], J. Bruggeman[3,4], E. Jeppesen[1,5], M.R. Flindt[2], L. van Gerven[6,7], J.H. Janse[6,8], A.B.G. Janssen[6,7], J.J. Kuiper[6,7], W.M. Mooij[6,7], D. Trolle[1,5]**

[1] Aarhus University, Department of Bioscience, Vejlsøvej 25, 8600 Silkeborg, Denmark

[2] University of Southern Denmark, Department of Biology, Campusvej 55, 5230 Odense M, Denmark

[3] Bolding & Bruggeman ApS, Strandgyden 25, 5466 Asperup, Denmark

[4] Plymouth Marine Laboratory, Prospect Place, The Hoe, Plymouth PL1 3DH, United Kingdom

[5] Sino-Danish Center for Education and Research, University of Chinese Academy of Sciences, Beijing

[6] Netherlands Institute of Ecology, Department of Aquatic Ecology, 6700 AB Wageningen, The Netherlands

[7] Wageningen University, Department of Aquatic Ecology and Water Quality Management, 6700 AA, The Netherlands

[8] PBL Netherlands Environmental Assessment Agency, Dept. of Nature and Rural Areas, 3720 AH Bilthoven, The Netherlands

*Corresponding author: Fenjuan Hu (fenjuan.hu@gmail.com)

## 1   Abstract

This study presents FABM-PCLake, a redesigned structure of the PCLake aquatic ecosystem model, which we implemented into the Framework for Aquatic Biogeochemical Models (FABM). In contrast to the original model, which was designed for temperate, fully mixed freshwater lakes, the new FABM-PCLake represents an integrated aquatic ecosystem model that can be linked with different hydrodynamic models and allows simulations of hydrodynamic and biogeochemical processes for zero-dimensional, one-dimensional as well as three-dimensional environments. FABM-PCLake describes interactions between multiple trophic levels, including piscivorous, zooplanktivorous and benthivorous fish, zooplankton, zoobenthos, three groups of phytoplankton and rooted macrophytes. The model also accounts for oxygen dynamics and nutrient cycling for nitrogen, phosphorus and silicon, both within the pelagic and benthic domains. FABM-PCLake includes a two-way communication between the biogeochemical processes and the physics, where some biogeochemical state variables (e.g., phytoplankton) influence light attenuation and thereby the spatial and temporal distributions of light and heat. At the same time, the physical environment, including water currents, light and temperature influence a wide range of biogeochemical processes. The model enables studies on ecosystem dynamics in physically heterogeneous environments (e.g., stratifying water bodies, and water bodies with horizontal gradient in physical and biogeochemical properties), and through FABM also enables data assimilation and multi-model ensemble simulations. Examples of potential new model applications include climate change impact studies and environmental impact assessment scenarios for temperate, sub-tropical and tropical lakes and reservoirs.

## 1   Introduction

The field of aquatic ecosystem modelling has undergone waves of development during the past decades, and models have grown in complexity in terms of ecosystem components and processes included (Robson, 2014). However, even though hundreds of models have been formulated for research or management purposes, only a handful has found frequent use and ongoing development (Trolle et al., 2012). This reflects that many models are being built with the same or similar properties, and thus that model development for the past decades has been subject to some degree of "re-inventing the wheel" as discussed by Mooij et al (2010). Another drawback of many aquatic ecosystem models is the typical discrepancy in

complexity between the ecosystem representation and the physical environment. High complexity in ecosystem conceptualizations therefore generally comes at the expense of simple or no hydrodynamic representation (e.g., PCLake (Janse and van Liere,1995; Janse, 2005; Janse et al., 2008) and EcoPath (Christensen and Pauly, 1992)). By contrast, physically resolved hydrodynamic models often include no or only simple ecosystem representations, and disregard higher trophic levels. Few studies have attempted to couple aquatic ecosystem dynamics with hydrodynamics (e.g., Hamilton and Schladow, 1997; Pereira et al., 2006; Fragoso et al., 2009), sometimes also including higher trophic levels (Makler-Pick et al., 2011). However, none of these models are validated for higher trophic levels (i.e. fish) and the source codes are also not readily available for further development. To avoid "re-inventing the wheel", and to overcome this discrepancy in complexity between the ecological and physical representation, a way forward is to enable an easy coupling between existing ecosystem models and hydrodynamic models. Thus, the complexity of the conceptual biogeochemical model and the physical representation may ideally easily be adapted to best suit the needs and purposes of a given study. Meanwhile, utilizing an open source platform would help promote model availability and also further development (Trolle et al., 2012). To this end, we implemented and modified a well-developed and widely applied ecosystem model, PCLake, within FABM, the Framework for Aquatic Biogeochemical Models by Bruggeman and Bolding (2014). FABM enables a flexible coupling of ecosystem processes in PCLake with a selection of hydrodynamic models representing zero- to three-dimensional hydrodynamics.

## 2   Implementation of PCLake in FABM

PCLake is originally a zero-dimensional ecological model for shallow lakes developed by Janse and van Liere (1995) and it has been widely applied (for example, Stonevičius and Taminskas, 2007; Mooij et al., 2009; Nielsen et al., 2014; further references in Mooij et al., 2010). The model describes the dynamics of phytoplankton, macrophytes and a simplified food web including zooplankton, zoobenthos, zooplanktivorous fish, benthivorous fish and piscivorous fish, and accounts for mass balances, represented by dry weight, nitrogen, phosphorus and silicon cycling between the various components of the ecosystem. The original PCLake model (documented in detail in Janse (2005)) contains detailed biological processes within the water column and also a relatively advanced biogeochemical sediment

module (describing nutrient dynamics in the sediment top layer and exchanges with the water
column), while thermo- and hydrodynamics are not explicitly accounted for. The original
model also includes a marsh module describing (helophytic) marsh vegetation in a zone
around a lake, which attempts to account for interactions between open waters and a more
highly vegetated marsh area that may be present close to the shoreline of some lakes. The
main purpose of the model is to predict critical nutrient loadings, i.e. the loading where a
shallow lake may switch between a clear and a turbid state, related to a non-linear ecosystem
response to nutrient loading as a result of self-enhancing feedback mechanisms within the
ecosystem.
FABM, in which we have now implemented PCLake, is a framework for biogeochemical
models of marine and freshwater systems (Bruggeman and Bolding, 2014). FABM enables
complex biogeochemical models to be developed as sets of stand-alone, process-specific
modules. These can be combined at runtime to create custom-tailored models. As outlined in
detail by Bruggeman and Bolding (2014), FABM divides the coupled advection-diffusion-
reaction equation that governs the dynamics of biogeochemical variables into two parts: a
reaction part (i.e., sink and source terms) provided by the biogeochemical models, and a
transport part handled by the hydrodynamic (i.e., physical) models. The transport part
includes advection, diffusion and potential vertical movements (sinking, floating and
potentially active movement), and also dilution and concentration processes. Therefore, based
on local variables (including, for example, local light conditions, temperature and
concentrations of state variables) provided by a hydrodynamic model, the biogeochemical
model calculate rates of sink and source terms at current time and space and pass the rates to
the hydrodynamic model via FABM. The hydrodynamic model will then handle numerical
integration of the biogeochemical processes and transport, and then pass updated states via
FABM back to the biogeochemical model – and this process will continue until the user-
defined end-time of a simulation. FABM thereby enables model applications of different
physical representations (ranging 0D to 3D) without the need to change the biogeochemical
source code. Most of the pelagic state variables in a biogeochemical model implemented in
FABM will typically be transported by the hydrodynamics. However, some pelagic variables,
particularly relevant for higher trophic levels such as fish (that may exhibit active movement,
based, for example, on the food source availability), can be set as exempt from hydrodynamic
transport or even include their own custom time and space varying movement. On the other
hand, all benthic state variables, such as macrophytes (that need to be attached to a "benthic"
grid cell), are always exempt from hydrodynamic transport. Further detail on the concept of
FABM is provided in Bruggeman and Bolding (2014).
Besides PCLake, a series of large ecosystem models have been implemented in FABM. These
include representations of the European Regional Seas Ecosystem Model (ERSEM,
Butenschön et al., 2016) and the lake model Aquatic EcoDynamics (AED, Hipsey et al.,
2013). But in contrast to PCLake, none of these include higher trophic levels such as fish.
FABM is written in Fortran2003 and therefore FABM-PCLake is also implemented in
Fortran2003. The key difference between the new FABM-PCLake (Fig. 1) and the original
PCLake conceptual model (e.g., Janse et al. 2010) is that FABM-PCLake can now be linked
to physical models. Hence, a major advantage of FABM-PCLake is that the detailed
biogeochemical processes provided by PCLake can now be used to study deep (i.e.
stratifying) and spatially complex aquatic ecosystems. While the core of the overall
conceptual model of the PCLake "lake part" remains intact, the underlying mechanisms of
processes that relate to transport have changed. For example, while the resuspension rate of
detritus (represented by an arrow going from the bottom sediments to the water column in
Fig. 1) is derived from an empirical relation to lake fetch in the original PCLake, resuspension
rate in FABM-PCLake can now be derived from the actual bottom shear stress as computed
by the physical model and passed via FABM to the biogeochemical model. When
implementing PCLake into FABM, a series of modifications relative to the original PCLake
model were made. This was done because some of the processes parameterized in the original
PCLake model can now be resolved explicitly by the hydrodynamic models and the
functionalities of FABM.
The main modifications are:
1) excluding the marsh module (as any two- or three-dimensional exchanges of solutes

25         can now be resolved by an explicit physical domain);

2) excluding the original loading, dilution and water level burial correction processes (as

27         this will now instead be resolved by the physical model and its boundary conditions);

3) excluding the original (and optional) forcing for dredging processes and fish

29         harvesting (as similar functionality is now provided through the state variable time

30         series forcing enabled by FABM);

4) adding the option to make resuspension directly dependent on bottom shear stress

32         provided by the hydrodynamic model. This functionality is derived from the PCLake

integral resuspension function and the shear-stress correlated resuspension function by Hamilton (1996), and may now be used as an alternative to the original empirical resuspension function, which was related only to the average lake fetch;

5) extending the available options for describing light limitation functions for individual phytoplankton groups and macrophytes (currently including both an integral function based on a Monod-type equation and the original Steele equation, which accounts for photo-inhibition (Di Toro and Matystik 1980)).

To maintain the integrity of the original PCLake model, in terms of process rates that are formulated using daily averaged incoming light, we used the ability of FABM to provide daily averaged values of photosynthetically active radiation (PAR) for the centre point in any given water column cell. In total, the FABM-PCLake implementation comprises 57 state variables. These include representations of oxygen dynamics, organic and inorganic forms of nitrogen, phosphorus and silicon, three phytoplankton groups, one zooplankton and one zoobenthos group, zooplanktivorous and zoobenthivorous fish (representing juveniles and adult fish, respectively), piscivorous fish and submerged macrophytes (Fig. 1). A complete record of the partial differential equations for each state variable can be found in the Supplementary Material.

The code implementation involved a complete redesign and rewrite of the PCLake code into a FABM compliant modular structure (see Fig. 2 and Supplementary material, supplementary table S1), thus allowing FABM to acquire sink and source terms for each state variable differential equation, and pass these for numerical solution and transportation by a physical host model. By implementing the model in FABM, FABM-PCLake has acquired the modularized code structure as other biogeochemical models within FABM and one can now combine PCLake modules with other modules from different biogeochemical models available in FABM to suit different study purposes. For example, one can run the phytoplankton module from the AED model together with the zooplankton module from the PCLake model simply by registering dependencies between the two modules via FABM. Specifically, this would be done by pointing to the AED phytoplankton module and state variable under the "coupling" section of the PCLake zooplankton module in the fabm.yaml file(see supplementary material, chapter S3) , which request specification of a food source. When coupling different modules from different models, one has to be aware of the units of individual state variables, and application of conversion factors may be needed to ensure that

state variables are in corresponding currencies, when coupled at runtime via FABM. Another important FABM feature is the ability to undertake data assimilation at runtime, where simulated state variables can be "relaxed" to values of observations that are read-in during a simulation. Hereby, one can assimilate certain components (e.g., macrophyte or zooplankton) of the ecosystem with observation data, while simulating other parts of the ecosystem dynamically. The model code was divided into modules of abiotic, phytoplankton, macrophytes and food web dynamics. These modules were further sub-divided into water column (pelagic) and sediment (benthic) domains. Concurrently, we developed an auxiliary module for FABM-PCLake to handle the overall system processes. The overall system processes are the processes that typically influence several modules, and they include resuspension, sedimentation and burial. In PCLake, burial is included as a representation of the natural process of sediment accumulation, which is caused by excessive sedimentation (resuspension rate < sedimentation rate) of particles at the sediment-water interface. The "buried" material is then considered inactive in the sediment biogeochemical processes and excluded from the system.

## 3 Model verification

To ensure that all biogeochemical processes have been implemented correctly through the equations in FABM-PCLake, we verified the model by running a benchmark test case against the original PCLake model. Hence, we compared output from the original PCLake model (zero-dimensional, using the OSIRIS version, i.e. a C++ executable called from a Microsoft Excel shell) with that from FABM-PCLake model executed with a zero-dimensional driver. The models were applied with identical model initialization and parameterization, and the same forcing and boundary conditions in terms of inflow, water temperature, light and nutrient loads for a 5-year period. The initial values for state variables and model parameterization were taken from the original PCLake version, which has been calibrated using data from 43 European lakes (Janse et al., 2010), most of which were Dutch lakes, but also included a few lakes from Belgium, Poland and Ireland. To ensure comparability, we left the Marsh module in the original PCLake model turned off, and used the simple empirical resuspension function (this function remains as an optional function in the FABM-PCLake model, while we also implemented a bottom stress driven resuspension process) in the FABM-PCLake version. Moreover, for the purpose of the benchmark test, processes that are

not included in the new FABM-PCLake, such as water column burial correction, dredging and fish harvesting, were turned off in the original PCLake model. We found that there were only marginal differences between the outputs of the two model versions, which could be attributed to small differences in the numerical solvers of the models (Fig. 3). We therefore conclude that the new FABM-PCLake implementation provides corresponding representations of ecosystem dynamics, relative to the original PCLake model.

## 4   Model applicability, limitations and perspectives

The FABM-PCLake model is now able to run with a selection of hydrodynamic models (which can be simply selected by the user), covering, for example, zero-dimensional (included with the FABM source code), one-dimensional (e.g., the General Ocean Turbulence Model, GOTM – http://www.gotm.net, and the General Lake Model, GLM – http://aed.see.uwa.edu.au/research/models/GLM) as well as three-dimensional (e.g., the General Estuary Transport Model, GETM – www.getm.eu, Modular Ocean Model, MOM - http://mom-ocean.org and work in progress - Nucleus for European Modelling of the Ocean, NEMO http://www.nemo-ocean.eu, and The Unstructured Grid Finite Volume Community Ocean Model, FVCOM - http://fvcom.smast.umassd.edu/fvcom ) hydrodynamic models. A major advantage of this development is that the detailed ecological processes provided by PCLake can now be used to study deep and spatially complex aquatic ecosystems. For example, macrophytes was originally represented as a single value in $g/m^2$ for a zero-dimensional model, but is now able to colonize different depths, for example when coupled to a 1D hypsographic hydrodynamic model, which allows a more gradual shift in the ecological states more typical for real lakes, even when shallow (Jeppesen et al., 2007). In addition, it becomes possible to study the concept of critical nutrient loading for spatially heterogeneous aquatic systems. This is important because the concept of regime shifts in ecosystems is widely acknowledged in science and ecosystem management, while the effect of spatial heterogeneity on the occurrence of regime shifts is poorly understood (Janssen et al., 2014). Other key features enabled by FABM are:

1) the ability to replace one or several of the PCLake modules (e.g., phytoplankton) with that from another ecosystem model available through FABM (e.g., ERGOM, ERSEM or AED);

2) the ability to assimilate observation data for some state variables while others are left fully dynamic (e.g., one could assimilate macrophyte biomass data, and simulate the response of fish, zooplankton, phytoplankton etc.);

3) the ability to run multiple models in an ensemble (e.g., for inter-model comparisons).

As we have tried to maintain the overall integrity of the ecological model PCLake, some process descriptions may still be improved to allow a more conceptually correct ecosystem representation in a physically explicit context. For example, higher hydrodynamic resolutions (i.e., 1D, 2D and 3D domains), could now allow a more advanced description of the behavior of macrophytes and fish. One example could be implementation of a more advanced macrophyte module that could dynamically re-allocate macrophyte biomass across pelagic grids such as the work presented by Sachse et al. (2014). Other examples counts potential advances for the fish module, which could include active fish movement (e.g., through an individual-based model), or implementation of the foraging arena theory (Ahrens et al. 2012) as adopted in the ECOPATH model.

**5   Sample case simulation outputs**

Whether run as a zero-, one- or three-dimensional model application, the model executable will generate an output file of NetCDF format (*.nc), which can be opened and manipulated by a range of software packages (e.g, Matlab, IDL) and a range of free NetCDF viewers, such as PyNcView (http://sourceforge.net/projects/pyncview). The latter provides an easy-to-use graphical user interface (GUI) for creation of animations and publication-quality figures. Figure 4 demonstrates a screenshot of this interface features, with visualization of FABM-PCLake state variables in a 1D context. Output from a one-year case simulation of temperature and macrophyte depth profiles is shown in Figure 5. This output was produced by linking FABM-PCLake with the 1D GOTM model (including a hypsograph that describes the relationship between depth and sediment area) for a hypothetical temperate 20m deep lake (with default PCLake parameterization).

**Code availability**

The model can be compiled and executed on Windows, Linux, and Mac OS machines, and is open source and freely available under the GNU General Public License (GPL) version 2. Source code, executables, and test cases can be downloaded directly from http://fabm.net, or as git repositories (updated information on how to download the code from git repositories as well as compiling the code for different platforms is available from the FABM wiki at http://fabm.net/wiki). Contact persons for FABM-PCLake model: Fenjuan Hu (fenjuan@bios.au.dk), Dennis Trolle (trolle@bios.au.dk), Karsten Bolding (bolding@bios.au.dk). Contact persons for the original zero-dimensional PCLake model: Jan H. Janse (jan.janse@pbl.nl), Wolf. M. Mooij (w.mooij@nioo.knaw.nl).

## Acknowledgements

This study was supported by CLEAR (a Villum Kann Rasmussen Foundation, Centre of Excellence project), and MARS (Managing Aquatic ecosystems and water Resources under multiple Stress) funded under the 7th EU Framework Program, Theme 6 (Environment including Climate Change), Contract No.: 603378 (http://www.mars-project.eu). A.B.G. Janssen was funded by the Netherlands Organization for Scientific Research (NWO) project no. 842.00.009. J.J. Kuiper and L.van Gerven were funded by the Netherlands Foundation for Applied Water Research (STOWA) project no. 443237 and the Netherlands Environmental Assessment Agency (PBL).

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

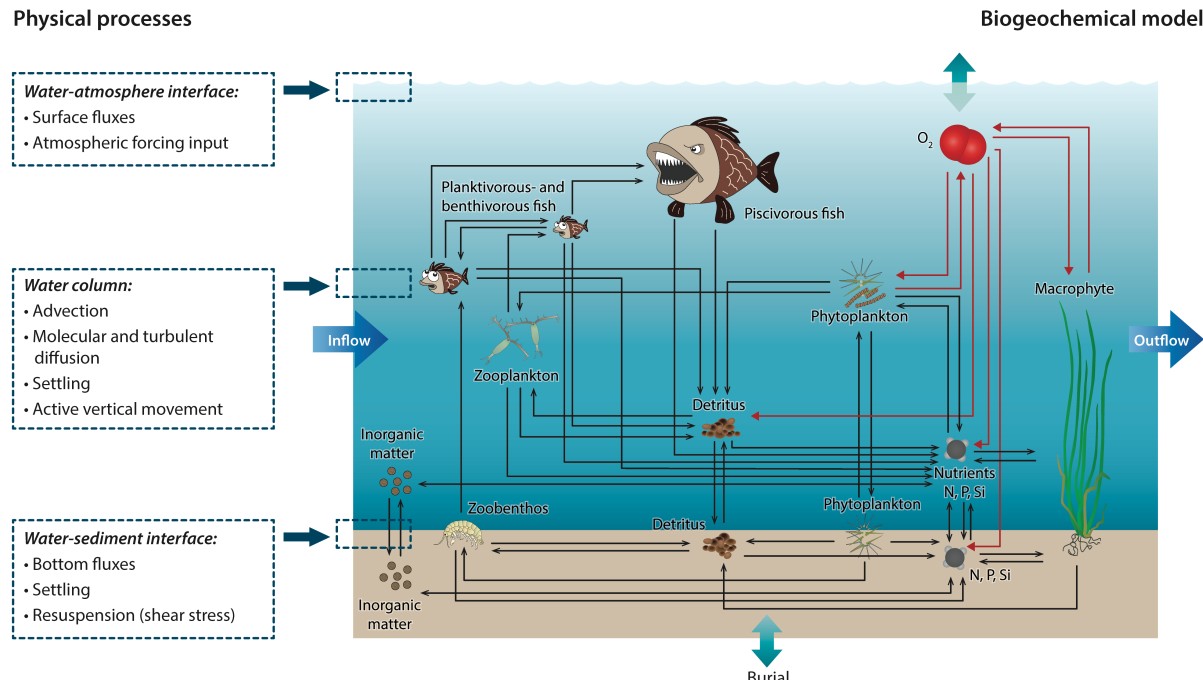

Fig. 1. Conceptual model of FABM-PCLake (FABM, Framework of Aquatic biogeochemical
Models; PCLake, the implemented aquatic ecosystem model). Key state variables of the
FABM-PCLake biogeochemical model and the interactions between these (represented by
arrows); and an indication of how a physical model may now transport biogeochemical state
variables through explicit physical processes.

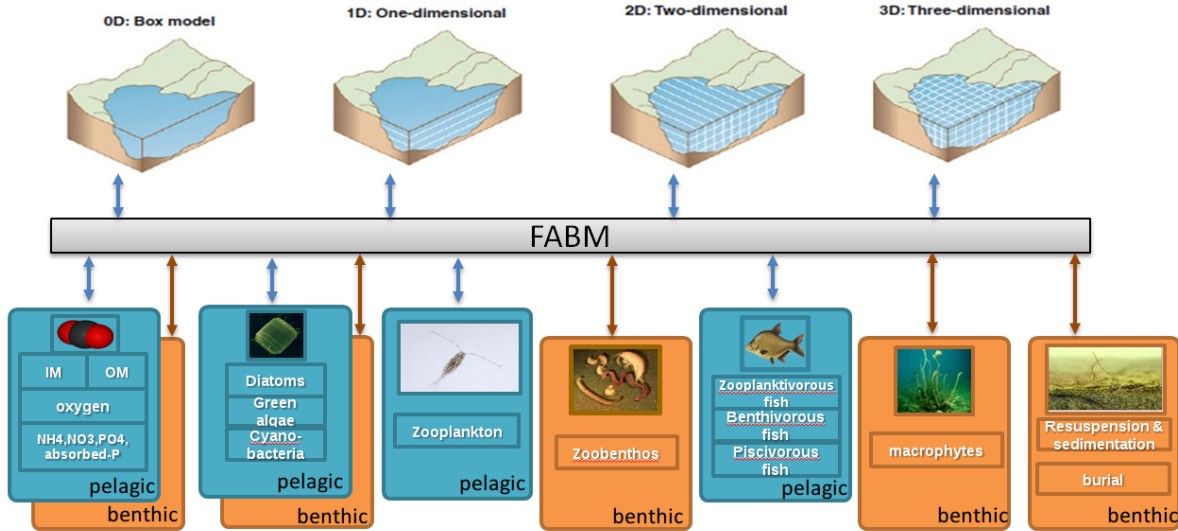

Fig. 2. The modular structure of the FABM-PCLake code. Each square box represents a
FORTRAN module of FABM-PCLake (and these modules are interacting/communicating
through FABM to simulate the processes illustrated by arrows in Fig.1). The brown coloured
boxes are related to the sediment domain and the blue boxes to the water column domain.
Note that all modules may be applied for 0-D to 3-D spatial domains. A detailed description
of the contents of each module is provided in the Supplementary Material.

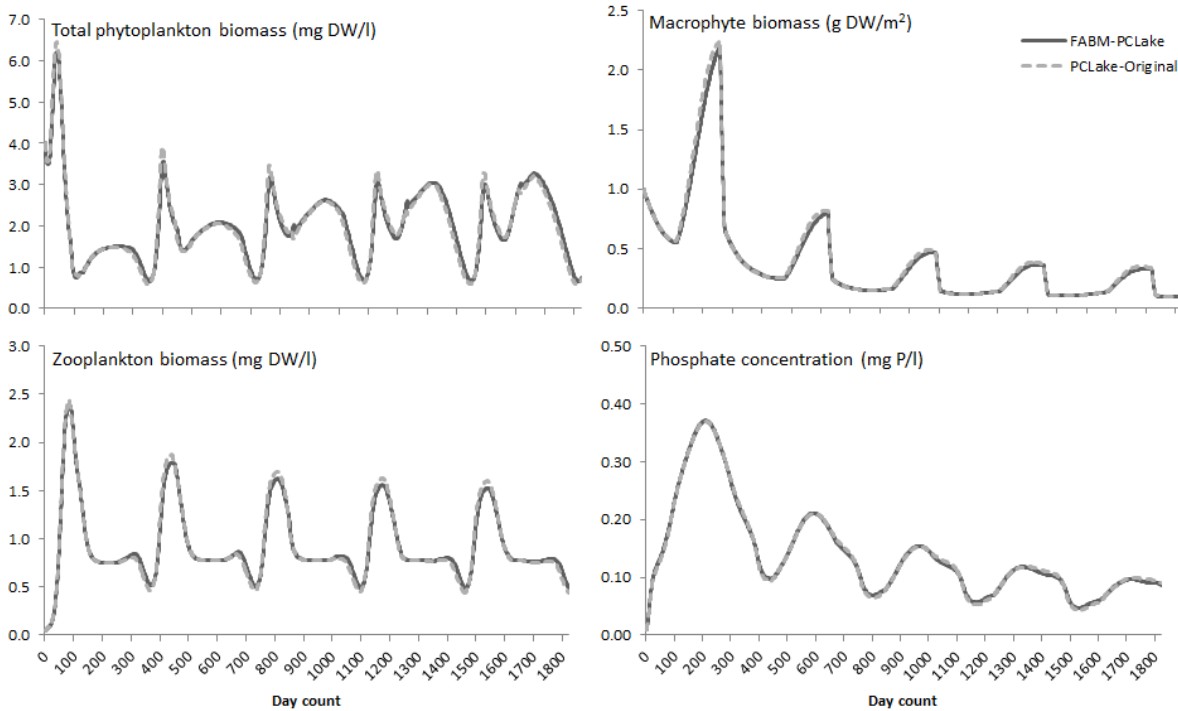

Fig. 3. Key time series outputs from a five-year simulation by the original PCLake model
(PCLake-Original), and the new FABM-PCLake model (FABM-PCLake), represented by dry
weight of total phytoplankton biomass, dry weight of zooplankton biomass, dry weight of
macrophytes biomass, and the concentration of phosphate in the water column.

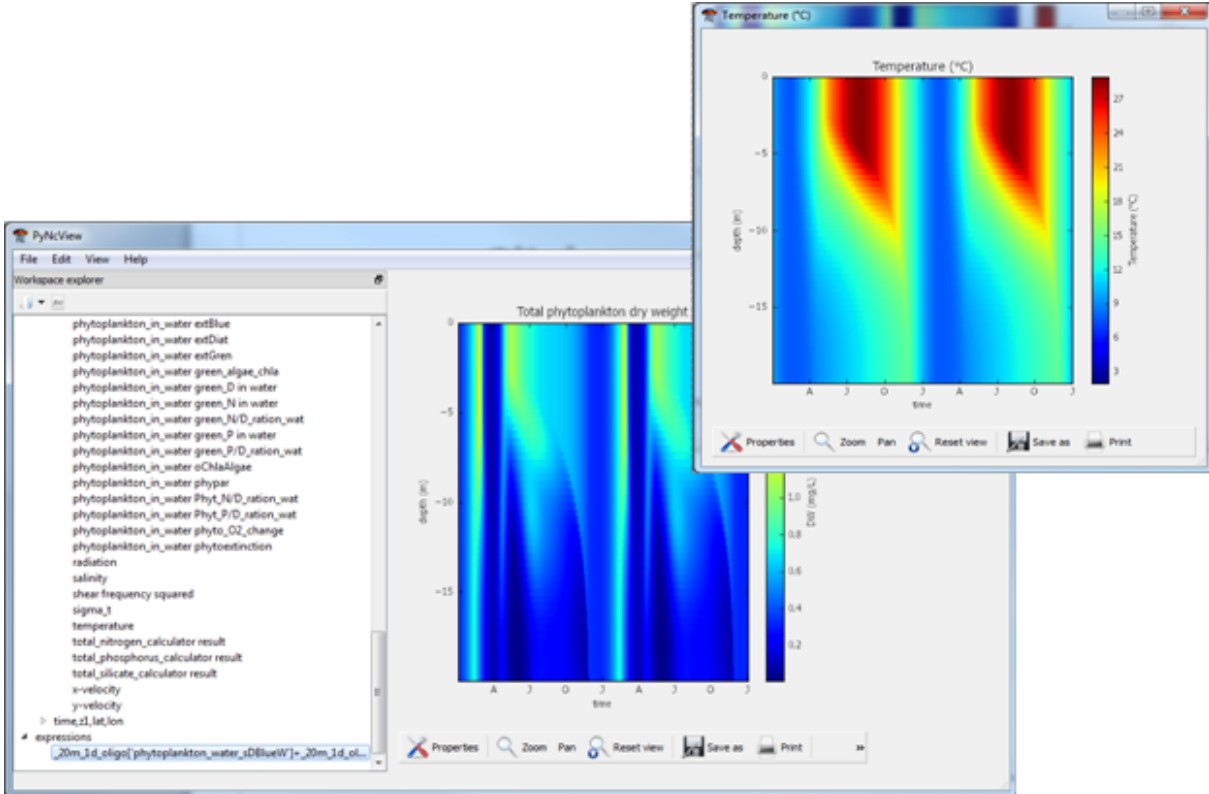

Fig. 4. Visualization of FABM-PCLake state variables in PyNcView, exemplified by a two-
year period simulated by a one-dimensional FABM-PCLake application of a 20 m deep water
column. State variables to be viewed are simply selected in the left panel, and figures can be
viewed, manipulated and saved in the right panel and as detached figures (a detached figure is
exemplified by the temperature plot).

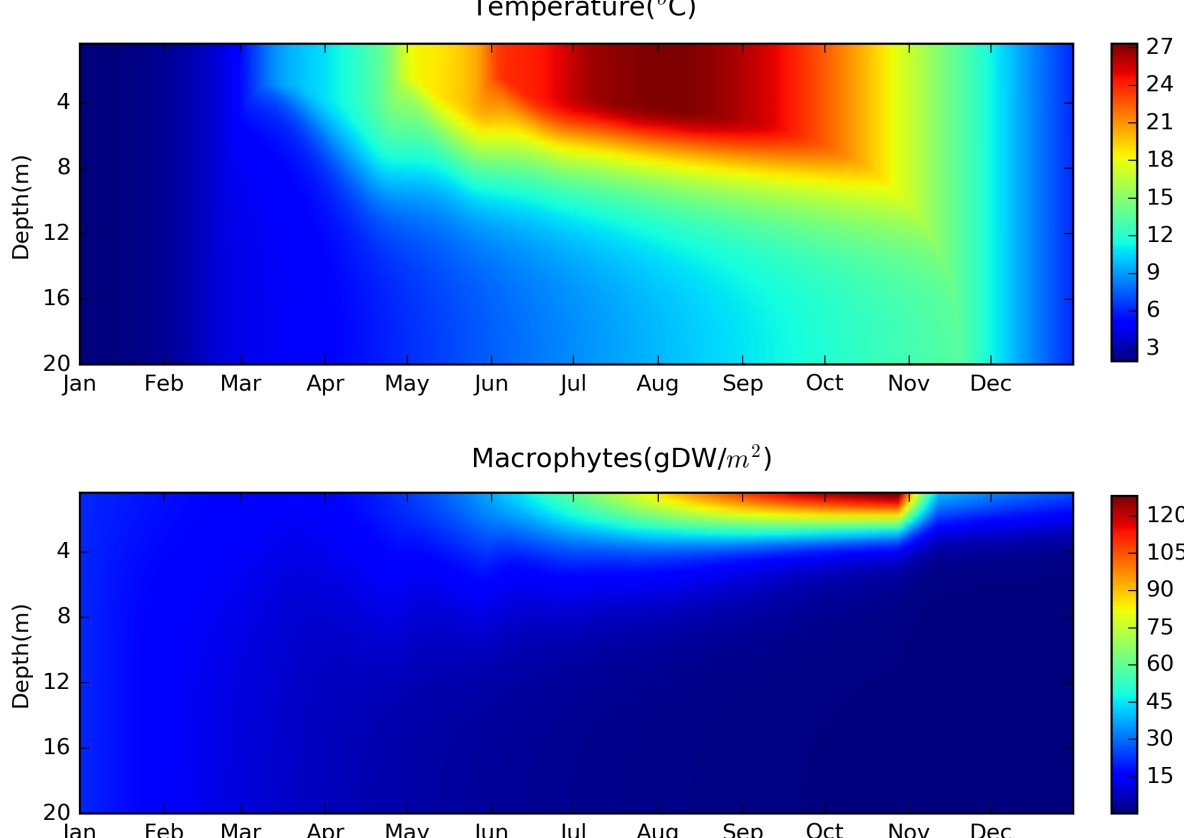

Fig. 5. Example of a one year simulation of temperature and macrophyte profiles  based on FABM-PCLake coupled to the 1D model GOTM (with hypsography enabled, meaning that each water column layer interfaces with a certain sediment area).