# Peer review of "FABM-PCLake – linking aquatic ecology with"

_Geoscientific Model Development, 2015_

## Referee Comment (RC1) · K. Rinke (Referee) · 1 Mar 2016

Review to gmd-2015-260 entitled "FABM-PCLake – linking aquatic ecology with hydro-dynamics by Fu et al

General comments

This manuscript is a concise description of the connection of PClake to FABM. This is a very powerful advancement and certainly of interest for the readership of GMD. The PClake-model is an ecological model of shallow lakes frequently used in science and management. Since shallow lakes are classically viewed as unstratified (ie mixed) systems, PClake is originally a 0D model, i.e. it models the lake as one mixed box. By connecting this model, which contained a high and reliable amount of ecological processes, to FABM it can also be applied outside the 0D context and coupled with

physical models for 1D or 3D hydrodynamics. This is a real step forward and certainly deserves publication. The MS is well written and well understandable. I have not seen major flaws or mistakes but I think that the scientific content of the paper could be improved. In its current form, the MS reads more like a newsgroup contribution and not yet like a full research paper. This would be different if 1-2 more applications would be shown. I also have a few questions about the details/consequences of the coupling and a few very minor points (see below).

I want to point out that I highly valued the excellent supplement material of this paper, which provides the experienced modeller with very helpful knowledge and facts.

Specific (major) comments

1. The paper would be improved if the abilities of the model would be shown with 1-2 more applications. This could be a simplified setting of a 3D model or a comparison of a 0D and a 1D simulation for a given system or a comparison of observed and modelled 1D dynamics in a given lake.... It would just leave a clearer impression of the abilities of the model system, particularly to those readers that have not yet heard from FABM et al

2. In the classical shallow lakes paradigm one has either a state with the dominance of pelagic primary producers (algae) or a state with the dominance of benthic primary producers (macrophytes). As soon as you move the PClake-model in a 1D-setting, every depth layer gets an own sediment surface attributed to this layer (derived from the bathymetric map). I expect that this spatial representation affects the competition between benthic and pelagic primary producers – the shallower a given 1D layer is located within the water column, the more superior becomes the benthic primary producer (because it gets more light). It is not clear to me, how exactly the benthic and pelagic compartments interact in a 1D setting – does each layer indeed have two separate ecological compartments (pelagic vs benthic?)? In the original 0D-setting everything is simple and clear because the benthic compartment is on the lake bottom

and the pelagic compartment on top of this. In the 1D setting, benthic and pelagic compartments coexist side by side within the same layer? Does that mean that the algae can never fully exclude macrophytes from the lake because the macrophytes can persist in the benthic compartments of the shallow layers (which may be a realistic condition?)??? Anyway – please explain in more detail. And keep in mind that this may become even more complex in a 3D-setting.

3. In a real lake, macrophytes grow into the pelagic compartment and can even fully occupy the pelagic compartment. Is it in the 1D setting allowed, that macrophytes can grow from there (home-)benthic layer upward into the next (pelagic) one above?

Minor comments

P2L27 has –> have

P4L16-19: I think in this setting the numerical integration of the ODE/PDEs is done by the physical model. This may be mentioned, if true

P4L29has –> have

P7L22-25: I am not sure whether I correctly understood the two final sentences about the sediment burial. Can you reformulate them?

———————————————

---

## Referee Comment (RC2) · Anonymous Referee #2 · 8 May 2016

This paper announces through a brief communication the coupling of PCLake to a variety of hydrodynamic models of various spatial representations (0D, 1D, 3D) via the Framework for Aquatic Biogeochemical Models (FABM). Of particular significance is (1) fully coupled linkage and feedback between PCLake and the physical model, (2) open source code with supporting contact persons, (3) compilation of code using Public License software, and (4) tailored output modules for comprehensive visualization.

The abstract states that the study involves a complete redesign of the PCLake model , but I would argue this is not the case and that the changes to the internal structure of of PCLake are largely incremental (e.g., sediment resuspension representation). The link to the hydrodynamic models is, however, a "redesign". I regard announcement of the new PCLake-FABM code as important and the paper fits within the scope of material of interest to the readers of GMDD. My only other major comments are that it

would be useful to have references associated with the new model developments (p. 5) including sediment resuspension and the additional options to describe light limitation of phytoplankton.

Minor comments: 1) The abstract has repetition; the physical models for heterogeneous environments (l. 7, l. 17). 2) The term "worldwide" at the bottom of the abstract is very open. A better specification would be useful. 3) Coupled 1D models of physics and water quality have been around for 2-3 decades – the text on p. 3, l. 4 could be more specific that few, if any, coupled models are actively used which have detailed representations of higher trophic level processes. 4) p. 4, l. 12: these physical processes are a subset of mixing and diffusion . 5) p. 4, ll. 14-23: the text here was confusing and requires revision. 6) p. 4, ll. 30-31: please associate references with the model. 7) p. 5, l. 3: FABM-PCLake can now be linked to physical process models. 8) p. 5: what is actually passed between the models; a shear stress from the physical model that enables material to be moved from bottom sediments to water? Fig. 1: indication rather than illustration? Could this diagram have something that really looked like a real fish? Fig. 2: I assume that phytoplankton are not restricted to these three groups? p. 5, l. 29: bases p. 5, l. 30: this description appears to imply that PAR in a cell is not depth integrated; the use of a centre point is not technically correct because of the exponential attenuation of light with depth. What is meant by the following: p. 6, l. 16: "enforce certain components" p. 6, l. 21: "overall system processes" p. 6, l. 23: "can prevent a net increase of sediment material". The latter relates to the fact that sediment accumulates naturally in all lakes, so some clarification is required. The comparison of the "old" and "new" PCLake results (Fig. 3) is impressive but I do not understand how they were almost identical with different resuspension Models? Wouldn't it had been easier to have switched off resuspension or was there calibration involved or did resuspension simply not occur? p. 7, l. 30: spatially p. 8, l. 8: "look at" = "simulate"

---

## Author Comment (AC1) · 13 May 2016

Dear Editor, reviewers and other interested members:

On behalf of the authors for the model description paper entitled "FABM-PCLake: linking aquatic ecology with hydrodynamics", we thank referees for dedicated and insightful comments. Please find our specific point-by-point responses to the referees" comments below. The original comments by the referees' are noted in default(black) color and our replies are provided in red color.

As part of the peer-review process, we have also improved the model code structure. This entailed further modularization of the code, which ultimately makes the model more user-friendly and also provides greater flexibility to adapt the conceptual model to individual systems. The code has been updated in the public repository. The code changes have not affected the biogeochemical processes of FABM-PCLake described here, but simply the way they are divided into different modules in the code. Please see the specific details provided under 'Additional revisions' at the end of this letter.

Please also refer to our supplement for tracked revision of the manuscript and updated supplementary materials.

Yours sincerely,

Fenjuan Hu

**Reply to Referee Nr.1**

*General Comments:*

This manuscript is a concise description of the connection of PClake to FABM. This is a very powerful advancement and certainly of interest for the readership of GMD. The PClake-model is an ecological model of shallow lakes frequently used in science and management. Since shallow lakes are classically viewed as stratified (i. e. mixed) systems, PClake is originally a 0D model, i.e. it models the lake as one mixed box. By connecting this model, which contained a high and reliable amount of ecological processes, to FABM it can also be applied outside the 0D context and coupled with physical models for 1D or 3D hydrodynamics. This is a real step forward and certainly deserves publication. The MS is well written and well understandable. I have not seen major flaws or mistakes but I think that the scientific content of the paper could be improved. In its current form, the MS reads more like a newsgroup contribution and not yet like a full research paper. This would be different if 1-2 more applications would be shown. I also have a few questions about the details/consequences of the coupling and a few very minor points (see below).

I want to point out that I highly valued the excellent supplement material of this paper, which provides the experienced modeller with very helpful knowledge and facts.

Reply: Thank you for the comprehensive summary of our work and the positive feedback relating to the supplementary material. In terms of the form (i.e. style) of the paper, we have followed the guidelines of GMD's "Model description papers". This include aspects relating to the scientific basis and purpose, the technical details of the model implementation (i.e. overview the numerical solution and the modular structure design), model verification (benchmark test), main model features as well as perspectives in relation to applications and further developments. We have included a new model application example, as suggested, which demonstrate a one year simulation of temperature and macrophyte profiles as simulated by FABM-PCLake coupled with the one-dimensional physical model GOTM, which include a hypsographic representation of the sediment-water interface (Fig.5). This plot also relates to the reviewers' comments' 1, 2 and 3, which we comment further on below.

**Major comments:**

1. The paper would be improved if the abilities of the model would be shown with 1-2 more applications. This could be a simplified setting of a 3D model or a comparison of a 0D and a 1D simulation for a given system or a comparison of observed and modeled 1D dynamics in a given lake.... It would just leave a clearer impression of the abilities of the model system, particularly to those readers that have not yet heard from FABM et al

Reply: As an additional example and manifestation of the models abilities, we have now included a new model application example, as suggested, which demonstrate a one year simulation of temperature and macrophyte profiles as simulated by FABM-PCLake coupled with the one-dimensional physical model GOTM. Output from the model application example is demonstrated in the new Fig. 5. We also provide details relating to the concept of the sediment-water interface of this 1D application, which also relate to comment 2 and 3 below.

2. In the classical shallow lakes paradigm one has either a state with the dominance of pelagic primary producers (algae) or a state with the dominance of benthic primary producers (macrophytes). As soon as you move the PClake-model in a 1D-setting, every depth layer gets an own sediment surface attributed to this layer (derived from the bathymetric map). I expect that this spatial representation affects the competition between benthic and pelagic primary producers – the shallower a given 1D layer is located within the water column, the more superior becomes the benthic primary producer (because it gets more light). It is not clear to me, how exactly the benthic and pelagic compartments interact in a 1D setting – does each layer indeed have two separate ecological compartments (pelagic vs benthic?)? In the original 0D-setting ev-

[Figure]

erything is simple and clear because the benthic compartment is on the lake bottom and the pelagic compartment on top of this. In the 1D setting, benthic and pelagic compartments coexist side by side within the same layer? Does that mean that the algae can never fully exclude macrophytes from the lake because the macrophytes can persist in the benthic compartments of the shallow layers (which may be a realistic condition?)??? Anyway – please explain in more detail. And keep in mind that this may become even more complex in a 3D-setting.

Reply: FABM-PCLake will function with any of the physical models for which a FABM interface has been written. This currently includes 0D, 1D as well as 3D models. The benthic-pelagic coupling is defined by the design of this interface. In practice, this means that as a model user, one can choose which physical representation is most suitable for the application purpose. For example, one can run FABM-PCLake with a standard GOTM (1D) set up, which means that only the most bottom layer will have a sediment-water interface. Alternatively, one can run FABM-PCLake with GOTM (1D) using a hypsographic representation of the sediment-water interface (simply by "turning ON" a hypsograph feature in GOTM), meaning that each individual water column layer is coupled to a certain sediment area (the size of this area relates to the hypsograph of the lake, which must then be provided as an input by the user). A model user may also choose a 3D model (e.g. GETM), which will then represent sediment-water interfaces for the bottommost cells in a three-dimensional domain. Hence, benthic and pelagic compartments can interact in both 1D and 3D settings, and in practice you would see that macrophytes may be present in the uppermost layers in a 1D model (with hypsograph), while being absent from deeper layers. To demonstrate this point, we have included an additional model application example (Fig. 5), where FABM-PCLake is coupled to GOTM using a hypsographic representation. Here, macrophytes are present in the uppermost layers (in this example extending to 4-5 meters depth), while at the same time being absent from deeper layers (where, for example, light conditions may not suffice for growth). The depth extent for macrophytes is influenced by light availability, temperature and nutrient concentrations in the sediments. Hence, macrophytes and phytoplankton in FABM-PCLake will compete for light, and if phyto-plankton concentration increases, the depth extent of macrophytes will decrease (or potentially be absent altogether). We think this is a more reliable description than the classical on-off (regime shift assumption) for shallow lakes. While on-off (clear-turbid) may be true for lakes that have a very flat bottom, most lakes have variable depth and show more gradual response to changes in loading such as the Danish lakes (see Jeppesen et al., 2007, for example).

3. In a real lake, macrophytes grow into the pelagic compartment and can even fully occupy the pelagic compartment. Is it in the 1D setting allowed, that macrophytes can grow from there (home-) benthic upward into the next (pelagic) one above?

Reply: Yes, the macrophytes in this case can extend into pelagic layer based on its height, and thereby influence light attenuation. The macrophyte code is at present similar to the original PCLake, meaning that the macrophyte height is specified simply by the user. In terms of perspectives relating to further code developments, it would be interesting to enable a dynamic macrophyte height simulation, e.g. similar to what has been presented by Sachse et al. (2014).

**Rply to referee Nr.2:**

*General Comments*

This paper announces through a brief communication the coupling of PCLake to a variety of hydrodynamic models of various spatial representations (0D, 1D, 3D) via the Framework for Aquatic Biogeochemical Models (FABM). Of particular significance is (1) fully coupled linkage and feedback between PCLake and the physical model, (2) open source code with supporting contact persons, (3) compilation of code using Public License software, and (4) tailored output modules for comprehensive visualization.

The abstract states that the study involves a complete redesign of the PCLake model, but I would argue this is not the case and that the changes to the internal structure of of PCLake are largely incremental (e.g., sediment resuspension representation). The link to the hydrodynamic models is, however, a "redesign". I regard announcement of the new PCLake-FABM code as important and the paper fits within the scope of material of interest to the readers of GMDD. My only other major comments are that it would be useful to have references associated with the new model developments (p. 5) including sediment resuspension and the additional options to describe light limitation of phytoplankton.

Reply: We have now changed the wording in the abstract to: "This study presents FABM-PCLake, a redesigned structure of the PCLake aquatic ecosystem model, which...". We have added additional references relating specifically to the background of the new features that we developed in relation to resuspension methods and light functions .

**Minor comments:**

1) The abstract has repetition; the physical models for heteroge- neous environments (l. 7, l. 17).

Reply: we have now changed the wording to avoid repetition on lines 7 band 17, respectively.

2) The term "worldwide" at the bottom of the abstract is very open. A better specification would be useful.

Reply: We have revised the text from "for lakes and reservoirs wordwide" to "for temperate, sub-tropical and tropical lakes and reservoirs".

3) Coupled 1D models of physics and water quality have been around for 2-3 decades – the text on p. 3, l. 4 could be more specific that few, if any, coupled models are actively used which have detailed representations of higher trophic level processes.

Reply: We have revised this sentence as followed, now including additional references:

"Few studies have attempted to couple aquatic ecosystem dynamics (e.g., Hamilton and Schladow, 1997; Pereira et al., 2006; Fragoso et al., 2009), sometimes also including higher trophic levels (Makler-Pick et al., 2011). However, none of these models are validated for higher trophic levels (i.e., fish) or readily available for further development."

4) p. 4, l. 12: these physical pro- cesses are a subset of mixing and diffusion .

Reply: The reviewer is correct, and we have now clarified this sentence (eddy-mixing is no longer mentioned explicitly).

5) p. 4, ll. 14-23: the text here was confusing and requires revision.

Reply: We have now tried to clarify the text in this particular section, which also refers to Bruggeman and Bolding 2014 for full details. The section now reads:

"Therefore, based on local variables (including, for example, local light conditions, temperature and concentrations of state variables) provided by a hydrodynamic model, the biogeochemical model calculate rates of sink and source terms at current time and space and pass the rates to the hydrodynamic model via FABM. The hydrodynamic model will then handle numerical integration of the biogeochemical processes and transport, and then pass updated states via FABM back to the biogeochemical model – and this process will continue until the user-defined end-time of a simulation."

6) p. 4, ll. 30-31: please associate references with the model.

Reply: We have added references for the specific models mentioned and updated the reference list accordingly.

7) p. 5, l. 3: FABM-PCLake can now be linked to physical process models.

Reply: Text has been changed accordingly.

8) p. 5: what is actually passed between the models; a shear stress from the physical model that enables material to be moved from bottom sediments to water? Fig. 1: indication rather than illustration? Could this diagram have something that really looked like a real fish?

Reply: Bottom shear stress is calculated by a physical model and then passed through FABM to the biogeochemical model. We have replaced the word "illustration" with "indication" in the caption for Fig. 1. We have revised this part with more caution in specifying these points, now reads as followed:

"For example, while the resuspension rate of detritus (represented by an arrow going from the bottom sediments to the water column in Fig. 1) is derived from an empirical relation to lake fetch in the original PCLake, resuspension rate in FABM-PCLake can now be derived from the actual bottom shear stress as computed by the physical model and passed via FABM to the biogeochemical model."

Fig. 2: I assume that phytoplankton are not restricted to these three groups?

Reply: The original PCLake model comprises three groups of phytoplankton (as depicted in the figure). This is also the standard configuration of FABM-PCLake. However, FABM allows coupling of individual biogeochemical models at runtime. Hence, a model user may simply configure a simulation to include none, one, two, three (etc.) FABM-

PCLake-phytoplankton modules. Thereby, the user has control of the complexity of the conceptual model and, for example, how many phytoplankton groups to include (and can also parameterize each phytoplankton group individually through input files, without the need to revise code).

p. 5, l. 29: bases p. 5, l. 30: this description appears to imply that PAR in a cell is not depth integrated; the use of a centre point is not technically correct because of the exponential attenuation of light with depth.

Reply: The text refers to how cell centre point PAR is passed between physical models and biogeochemical models – and not how these values are processed by light functions to derive primary production. There are multiple light functions implemented in biogeochcemical models in FABM; some utilize centre point PAR while others use the PAR value at top and bottom of a layer for deriving depth integrated PAR.

What is meant by the following: p. 6, l. 16: "enforce certain components" p. 6, l. 21: "overall system processes" p. 6, l. 23: "can prevent a net increase of sediment material". The latter relates to the fact that sediment accumulates naturally in all lakes, so some clarification is required.

Reply: We have revised this part, which now reads:

"The overall system processes are the processes that typically influence several other modules, and they include resuspension, sedimentation and burial. In PCLake, burial is included as a representation of the natural process of sediment accumulation, which is caused by excessive sedimentation (resuspension rate < sedimentation rate) of particles at the sediment-water interface. The "buried" material is then considered inactive in the sediment biogeochemical processes and excluded from the system."

The comparison of the "old" and "new" PCLake results (Fig. 3) is impressive but I do not understand how they were almost identical with different resuspension Models? Wouldn't it had been easier to have switched off resuspension or was there calibration involved or did resuspension simply not occur?

Reply: This is a benchmark test, for which the main purpose is to test that the new model can produce identical output relative to the old model. Therefore, and as specified on p. 7, both the 'old' and 'new' PCLake model simulation make use of the same (old) empirical resuspension function for this comparison.

p. 7, l. 30: spatially p. 8, l. 8: "look at" = "simulate"

Reply: We have changed text accordingly.

**Additional revisions since original submission:**

*Revision of FABM-PCLake code and module structure*

To be completely consistent with the modular design philosophy of FABM, the FABM-PCLake's original foodweb-water module have been separated into a zooplankton and fish module, respectively. Accordingly, foodweb-sediment module has been renamed as zoobenthos. This separation enables greater flexibility when designing the conceptual model for a specific system as modules may be turned On or Off, repeated several times (e.g. to include multiple zooplankton groups rather than just a single group) or replaced by another biogeochemical module available within FABM. The names of the modules and source files have been changed accordingly. The source files, the Supplementary Material and Fig. 2 have been updated accordingly.

**References:**

[revised manuscript text omitted]

---

## Referee Report (RR1)

Review of GMD-2015-260 entitled "FABM-PCLake – linking aquatic ecology with hydrodynamics"

Thank you very much for the revised version of the MS. I like the case study simulation that you added to your manuscript and think that the current manuscript fulfills the requirements for being published in GMD. I have only one small question remaining and would be happy if you could comment on that in the text:

You are stating that it is possible to combine the application of PClake within FABM with selected submodules from other models available under FABM, for example:

P6L24-26: ...such as running the phytoplankton module from the AED model together with the zooplankton module from the PCLake model to simulate the ecosystem for a particular case study

A comparable statement is given at P8L22-24. I would like to know a bit more how exactly this would take place.

You correctly mentioned that running PClake under FABM allows to let the model run with different physical drivers (eg GLM, GOTM, GETM) without requiring changes on the source code. Does this statement also holds true if I combine different submodules from different ecological models? To give an example: If I want to combine the phytoplankton module from PClake with the zooplankton from AED, how do the state variables from different models exchange the information without changing the source code? In other words, how can an AED-zooplankter "know" how much PClake-phytoplankton is available for grazing? And if grazing is taking place, how does PClake getting to "know" the quantum of algae taken away by the AED-zooplankter?

I do not expect a full technical guideline but ask for some additional information that illustrates the reader how this will be realised.

---

## Author Response (AR2)

Dear editor,
We have now revised our manuscript according to the minor text revision requested by the reviewer. Please see our response below.

Kind regards,
Fenjuan Hu

**Reviewer comment:**

*A comparable statement is given at P8L22-24. I would like to know a bit more how exactly this would take place.*
*You correctly mentioned that running PClake under FABM allows to let the model run with different physical drivers (eg GLM, GOTM, GETM) without requiring changes on the source code. Does this statement also holds true if I combine different submodules from different ecological models? To give an example: If I want to combine the phytoplankton module from PClake with the zooplankton from AED, how do the state variables from different models exchange the information without changing the source code? In other words, how can an AED-zooplankter "know" how much PClake-phytoplankton is available for grazing? And if grazing is taking place, how does PClake getting to "know" the quantum of algae taken away by the AED-zooplankter?*
*I do not expect a full technical guideline but ask for some additional information that illustrates the reader how this will be realised.*

**Authors reply:**

Yes, you can combine biogeochemical modules from different models at runtime without the need to change any code whatsoever. This is done simply through a FABM input file, which allows you to specify exactly which state variables, and from which modules, a specific module should retrieve and exchange data. For example, the zooplankton module of PCLake can retrieve phytoplankton as a food source from a AED phytoplankton module – and FABM will keep track of the transfer rates between these modules. We have added additional detail on how this is accomplished in the manuscript text (and more technical detail on this aspect is also available on fabm.net). This particular section now reads:

[revised manuscript text omitted]